# Diagonal State Spaces
# are as Effective as Structured State Spaces

**Ankit Gupta***
IBM Research
ankitgupta.iitkanpur@gmail.com

**Albert Gu**
Stanford University
albertgu@stanford.edu

**Jonathan Berant**
Tel Aviv University
joberant@cs.tau.ac.il

## Abstract

Modeling long range dependencies in sequential data is a fundamental step towards attaining human-level performance in many modalities such as text, vision, audio and video. While attention-based models are a popular and effective choice in modeling short-range interactions, their performance on tasks requiring long range reasoning has been largely inadequate. In an exciting result, Gu et al. (2022) proposed the *Structured State Space* (S4) architecture delivering large gains over state-of-the-art models on several long-range tasks across various modalities. The core proposition of S4 is the parameterization of state matrices via a diagonal plus low rank structure, allowing efficient computation. In this work, we show that one can match the performance of S4 even without the low rank correction and thus assuming the state matrices to be diagonal. Our *Diagonal State Space (DSS)* model matches the performance of S4 on Long Range Arena tasks, speech classification on Speech Commands dataset, while being conceptually simpler and straightforward to implement.

## 1 Introduction

The Transformer architecture [VSP+17] has been successful across many areas of machine learning. Transformers pre-trained on large amounts of unlabelled text via a denoising objective have become the standard in natural language processing, exhibiting impressive amounts of linguistic and world knowledge [BMR+20, BMH+22]. This recipe has also led to remarkable developments in the areas of vision [RPG+21, RKH+21] and speech [DXX18, BZMA20].

The contextualizing component of the Transformer is the multi-head attention layer which, for inputs of length $L$, has an expensive $\Omega(L^2)$ complexity. This becomes prohibitive on tasks where the model is required to capture long-range interactions of various parts of a long input. To alleviate this issue, several Transformer variants have been proposed with reduced compute and memory requirements [QML+20, KKL20, RSVG21, BPC20, GB20, KVPF20, WLK+20, CLD+21, ZS21, GDG+21] (see [TDBM20] for a survey). Despite this effort, all these models have reported inadequate performance on benchmarks created to formally evaluate and quantify a model's ability to perform long-range reasoning (such as Long Range Arena [TDA+21] and SCROLLS [SSI+22]).

In a recent breakthrough result, Gu et al. [GGR22] proposed S4, a sequence-to-sequence model that uses linear state spaces for contextualization instead of attention. It has shown remarkable performance on tasks requiring long-range reasoning in domains such as text, images and audio. For instance, on Long Range Arena it advances the state-of-the-art by 19 accuracy points over the best performing Transformer variant. Its remarkable abilities are not limited to text and images and carry over to tasks such as time-series forecasting, speech recognition and audio generation [GGDR22].

Despite S4's achievements, its design is complex and is centered around the HiPPO theory, which is a mathematical framework for long-range modeling [VKE19, GDE+20, GJG+21]. [GGR22]

---

*work done while author was part of IBM AI Residency program.

36th Conference on Neural Information Processing Systems (NeurIPS 2022).

showed that state space models with various alternative initializations perform poorly in comparison to initializing the state-space parameters with a particular HiPPO matrix. In order to leverage this matrix, they parameterize the learned state spaces using a Diagonal Plus Low Rank (DPLR) structure and, as a result, need to employ several reduction steps and linear algebraic techniques to be able to compute the state space output efficiently, making S4 difficult to understand, implement and analyze.

In this work, we show that it is possible to match S4's performance while using a much simpler, fully-diagonal parameterization of state spaces. While we confirm that random diagonal state spaces are less effective, we observe that there do in fact exist effective diagonal state matrices: simply removing the low-rank component of the DPLR HiPPO matrix still preserves its performance. Leveraging this idea, our proposed Diagonal State Space (DSS) model enforces state matrices to be diagonal, making it significantly simpler to formulate, implement and analyze, while being provably as expressive as general state spaces. In contrast to S4, DSS does not assume any specialized background beyond basic linear algebra and can be implemented in just a few lines of code. Our implementation fits in a single page and is provided in §A.7 (Figure 6).

We evaluate the performance of DSS on Long Range Arena (LRA) which is a suite of sequence-level classification tasks with diverse input lengths ($1K$-$16K$) requiring similarity, structural, and visual-spatial reasoning over a wide range of modalities such as text, natural/synthetic images, and mathematical expressions. Despite its simplicity, DSS delivers an average accuracy of $81.88$ across the 6 tasks of LRA, comparable to the state-of-the-art performance of S4 ($80.21$). In addition, DSS maintains a comfortable 20 point lead over the best Transformer variant ($81.88$ vs $61.41$).

In the audio domain, we evaluate the performance of DSS on raw speech classification. On the Speech Commands dataset [War18], which consists of raw audio samples of length $16K$, we again found the performance of DSS to be comparable to that of S4 ($98.2$ vs $98.1$).

To summarize, our results demonstrate that DSS is a simple and effective method for modeling long-range interactions in modalities such as text, images and audio. We believe that the effectiveness, efficiency and transparency of DSS can significantly contribute to the adoption of state space models over their attention-based peers. Our code is available at `https://github.com/ag1988/dss`.

## 2 Background

We start by reviewing the basics of time-invariant linear state spaces.

**State Spaces** A continuous-time state space model (SSM) parameterized by the state matrix $A \in \mathbb{R}^{N \times N}$ and vectors $B \in \mathbb{R}^{N \times 1}$, $C \in \mathbb{R}^{1 \times N}$ is given by the differential equation:

$$\frac{dx}{dt}(t) = Ax(t) + Bu(t) \ , \ y(t) = Cx(t) \tag{1}$$

which defines a function-to-function map $u(t) \mapsto y(t)$. For a given value of time $t \in \mathbb{R}$, $u(t) \in \mathbb{R}$ denotes the value of the input signal $u$, $x(t) \in \mathbb{R}^{N \times 1}$ denotes the state vector and $y(t) \in \mathbb{R}$ denotes the value of the output signal $y$. We call a state space *diagonal* if it has a diagonal state matrix.

**Discretization** For a given sample time $\Delta \in \mathbb{R}_{>0}$, the discretization of a continuous state space (Equation 1) assuming *zero-order hold*[2] on $u$ is defined as a sequence-to-sequence map from $(u_0, \ldots, u_{L-1}) = u \in \mathbb{R}^L$ to $(y_0, \ldots, y_{L-1}) = y \in \mathbb{R}^L$ via the recurrence,

$$x_k = \overline{A}x_{k-1} + \overline{B}u_k \ , \ y_k = \overline{C}x_k$$

$$\overline{A} = e^{A\Delta} \ , \ \overline{B} = (\overline{A} - I)A^{-1}B \ , \ \overline{C} = C \ . \tag{2}$$

Assuming $x_{-1} = 0$ for simplicity, this recurrence can be explicitly unrolled as

$$y_k \ = \ \sum_{j=0}^{k} \overline{C}\,\overline{A}^j\overline{B} \cdot u_{k-j} \ . \tag{3}$$

For convenience, the scalars $\overline{C}\,\overline{A}^k\overline{B}$ are gathered to define the SSM kernel $\overline{K} \in \mathbb{R}^L$ as

$$\overline{K} \ = \ (\overline{CB}, \overline{CAB}, \ldots, \overline{CA}^{L-1}\overline{B}) \ = \ (\, Ce^{A\cdot k\Delta}(e^{A\Delta} - I)A^{-1}B \,)_{0 \leqslant k < L}, \tag{4}$$

---

[2] assumes the value of a sample of $u$ is held constant for a duration of one sample interval $\Delta$ [Gaj03].

where the last equality follows by substituting the values of $\overline{A}, \overline{B}, \overline{C}$ from Equation 2. Hence,

$$y_k = \sum_{j=0}^{k} \overline{K}_j \cdot u_{k-j} .\tag{5}$$

Given an input sequence $u \in \mathbb{R}^L$, it is possible to compute the output $y \in \mathbb{R}^L$ sequentially via the recurrence in Equation 2. Unfortunately, sequential computation is prohibitively slow on long inputs and, instead, Equation 5 can be used to compute all elements of $y$ in parallel, provided we have already computed $\overline{K}$.

**Computing $y$ from $u$ and $\overline{K}$ is easy.** Given an input sequence $u \in \mathbb{R}^L$ and the SSM kernel $\overline{K} \in \mathbb{R}^L$, naively using Equation 5 for computing $y$ would require $O(L^2)$ multiplications. Fortunately, this can be done much more efficiently by observing that for the univariate polynomials

$$\overline{K}(z) = \sum_{i=0}^{L-1} \overline{K}_i z^i \text{ and } u(z) = \sum_{i=0}^{L-1} u_i z^i,$$

$y_k$ is the coefficient of $z^k$ in the polynomial $\overline{K}(z) \cdot u(z)$, i.e. all $y_k$'s can be computed simultaneously by multiplying two degree $L-1$ polynomials. It is well-known that this can be done in $O(L \log(L))$ time via Fast Fourier Transform (FFT) [CLRS09]. We denote this fast computation of Equation 5 via the discrete convolution as

$$y = \overline{K} * u .\tag{6}$$

Hence, given the SSM kernel $\overline{K} \in \mathbb{R}^L$, the output of a discretized state space can be computed efficiently from the input. The challenging part is computing $\overline{K}$ itself as it involves computing $L$ distinct matrix powers (Equation 4). Similar to S4, instead of directly using $A, B, C$, our idea is to use an alternate parameterization of state spaces for which it would be easier to compute $\overline{K}$.

# 3 Method

Having stated the necessary background, we now turn to the main contribution of our work.

## 3.1 Diagonal State Spaces

For a general state matrix $A$ it is expensive to compute the matrix exponentials $e^{A \cdot k \Delta}$ for all $0 \leqslant k < L$ (Equation 4). However, if $A$ was diagonalizable over $\mathbb{C}$ and we were given its eigendecomposition $A = V \Lambda V^{-1}$, where $\Lambda$ is diagonal, then we could easily compute $e^{A \cdot k \Delta} = V e^{\Lambda \cdot k \Delta} V^{-1}$ as it is trivial to exponentiate diagonal matrices.[3] In this case Equation 4 simplifies to

$$\overline{C} \overline{A}^k \overline{B} = C e^{V \Lambda V^{-1} \cdot k \Delta} (e^{V \Lambda V^{-1} \Delta} - I)(V \Lambda V^{-1})^{-1} B = (CV) e^{\Lambda k \Delta} (e^{\Lambda \Delta} - I) \Lambda^{-1} (V^{-1} B) .$$

Leveraging the above observation, we base our model on the following proposition which asserts that, under mild technical assumptions, diagonal state spaces are as expressive as general state spaces. We use the operator $\overline{K}_{\Delta,L}(A, B, C) \in \mathbb{C}^{1 \times L}$ to denote the kernel (Equation 4) of length $L$ for state space $(A, B, C)$ over $\mathbb{C}$ and sample time $\Delta$.

**Proposition 1.** *Let $K \in \mathbb{C}^{1 \times L}$ be the kernel of length $L$ of a given state space $(A, B, C)$ and sample time $\Delta > 0$, where $A \in \mathbb{C}^{N \times N}$ is diagonalizable over $\mathbb{C}$ with eigenvalues $\lambda_1, \ldots, \lambda_N$ and $\forall i, \lambda_i \neq 0$ and $e^{L \lambda_i \Delta} \neq 1$. Let $P \in \mathbb{C}^{N \times L}$ be $P_{i,k} = \lambda_i k \Delta$ and $\Lambda$ be the diagonal matrix with $\lambda_1, \ldots, \lambda_N$. Then there exist $\widetilde{w}, w \in \mathbb{C}^{1 \times N}$ such that*

*(a)* $K = \overline{K}_{\Delta,L}(\Lambda, (1)_{1 \leqslant i \leqslant N}, \widetilde{w}) = \widetilde{w} \cdot \Lambda^{-1}(e^{\Lambda \Delta} - I) \cdot \text{elementwise-exp}(P),$

*(b)* $K = \overline{K}_{\Delta,L}(\Lambda, ((e^{L \lambda_i \Delta} - 1)^{-1})_{1 \leqslant i \leqslant N}, w) = w \cdot \Lambda^{-1} \cdot \text{row-softmax}(P).$

---

[3] $e^A = \sum_{j=0}^{\infty} \frac{A^j}{j!} = \sum_{j=0}^{\infty} \frac{(V \Lambda V^{-1})^j}{j!} = V \left( \sum_{j=0}^{\infty} \frac{\Lambda^j}{j!} \right) V^{-1}$. For $\Lambda = \text{diag}(\lambda_1, \ldots, \lambda_N)$ we have $\Lambda^j = \text{diag}(\lambda_1^j, \ldots, \lambda_N^j)$ and hence $e^A = V \cdot \text{diag}(\exp(\lambda_1), \ldots, \exp(\lambda_N)) \cdot V^{-1}$.

| | |
|---|---|
| **Input:** parameters $\Lambda_{\mathrm{re}}, \Lambda_{\mathrm{im}} \in \mathbb{R}^N, w \in \mathbb{C}^N$, sample time $\Delta_{\log} \in \mathbb{R}$ | |
| **Output:** SSM kernel $\overline{K} \in \mathbb{R}^L$ (Proposition 1(b)) | |

| | |
|---|---|
| 1: $\Lambda \leftarrow \Lambda_{\mathrm{re}} + i \cdot \Lambda_{\mathrm{im}}$ , $\Delta \leftarrow \exp(\Delta_{\log})$ | $\triangleright \Delta$ is positive real |
| 2: $P_{N \times L} \leftarrow (\Delta * \Lambda)_{N \times 1} * [0, 1, \ldots L-1]_{1 \times L}$ | $\triangleright$ outer product |
| 3: $S \leftarrow \text{row-softmax}_\epsilon(P)$ | $\triangleright$ numerically stable (§A.2) |
| 4: $\overline{K} \leftarrow \mathrm{Re}(\,(w/\Lambda)_{1 \times N} \cdot S\,)$ | $\triangleright$ real part[5] |

Algorithm 1: DSS$_{\text{SOFTMAX}}$ KERNEL (SKETCH)

The proof of Proposition 1 is provided in §A.1. In the above equations, the last equality follows by using Equation 4 to explicitly compute the expression for the kernel of the corresponding diagonal state space. Hence, for any given state space with a well-behaved state matrix there exists a diagonal state space computing the same kernel.[4] More importantly, the expressions for the kernels of the said diagonal state spaces no longer involve matrix powers but only a structured matrix-vector product.

Proposition 1(a) suggests that we can parameterize state spaces via $\Lambda, \widetilde{w} \in \mathbb{C}^N$ and simply compute the kernel as shown in Proposition 1(a). Unfortunately, in practice, the real part of the elements of $\Lambda$ can become positive during training making the training unstable on long inputs. This is because the matrix elementwise-$\exp(P)$ contains terms as large as $\exp(\lambda_i \Delta(L-1))$ which even for a modest value of $L$ can be very large. To address this, we propose two methods to model diagonal state spaces.

**DSS$_{\text{EXP}}$** In this variant, we use Proposition 1(a) to model our state space but restrict the real part of elements of $\Lambda$ to be negative. DSS$_{\text{EXP}}$ has parameters $\Lambda_{\mathrm{re}}, \Lambda_{\mathrm{im}} \in \mathbb{R}^N, \widetilde{w} \in \mathbb{C}^N$ and $\Delta_{\log} \in \mathbb{R}$. $\Lambda$ is computed as $-\text{elementwise-}\exp(\Lambda_{\mathrm{re}}) + i \cdot \Lambda_{\mathrm{im}}$ where $i = \sqrt{-1}$ [GGDR22]. $\Delta$ is computed as $\exp(\Delta_{\log}) \in \mathbb{R}_{>0}$ and the kernel is then computed via Proposition 1(a).

DSS$_{\text{EXP}}$ provides a remarkably simple computation of state space kernels but restricts the space of the learned $\Lambda$ (the real part must be negative). It is not clear if such a restriction could be detrimental for some tasks, and we now present an alternate method that provides the simplicity of Proposition 1(a) while being provably as expressive as general state spaces.

**DSS$_{\text{SOFTMAX}}$** Instead of restricting the elements of $\Lambda$, another option for bounding the elements of elementwise-$\exp(P)$ is to normalize each row by the sum of its elements, which leads us to Proposition 1(b). DSS$_{\text{SOFTMAX}}$ has parameters $\Lambda_{\mathrm{re}}, \Lambda_{\mathrm{im}} \in \mathbb{R}^N, w \in \mathbb{C}^N$ and $\Delta_{\log} \in \mathbb{R}$. $\Lambda$ is computed as $\Lambda_{\mathrm{re}} + i \cdot \Lambda_{\mathrm{im}}$, $\Delta$ is computed as $\exp(\Delta_{\log}) \in \mathbb{R}_{>0}$, and the kernel is then computed via Proposition 1(b). A sketch of this computation is presented in Algorithm 1. We note that, unlike over $\mathbb{R}$, softmax can have singularities over $\mathbb{C}$ and slight care must be taken while computing it (e.g., softmax$(0, i\pi)$ is not defined). We instead use a corrected version softmax$_\epsilon$ and detail it in §A.2.

### 3.2 Diagonal State Space (DSS) Layer

We are now ready to describe the DSS layer. We retain the skeletal structure of S4 and simply replace the parameterization and computation of the SSM kernel by one of the methods described in §3.1.

Each DSS layer receives a length-$L$ sequence $u$ of $H$-dimensional vectors as input, i.e., $u \in \mathbb{R}^{H \times L}$, and produces an output $y \in \mathbb{R}^{H \times L}$. The parameters of the layer are $\Lambda_{\mathrm{re}}, \Lambda_{\mathrm{im}} \in \mathbb{R}^N, \Delta_{\log} \in \mathbb{R}^H$ and $W \in \mathbb{C}^{H \times N}$. For each coordinate $h = 1, \ldots, H$, a state space kernel $\overline{K}_h \in \mathbb{R}^L$ is computed using one of methods described in §3.1. For example, in case of DSS$_{\text{SOFTMAX}}$, Algorithm 1 is used with parameters $\Lambda_{\mathrm{re}}, \Lambda_{\mathrm{im}} \in \mathbb{R}^N, W_h \in \mathbb{C}^N$ and $(\Delta_{\log})_h \in \mathbb{R}$. The output $y_h \in \mathbb{R}^L$ for coordinate $h$ is computed from $u_h \in \mathbb{R}^L$ and $\overline{K}_h$ using Equation 6.

Following S4, we also add a residual connection from $u$ to $y$ followed by a GELU non-linearity [HG16]. This is followed by a position-wise linear projection $W_{\text{out}} \in \mathbb{R}^{H \times H}$ to enable information exchange among the $H$ coordinates.

---

[4] It is possible for norms of the parameters of the resulting diagonal state spaces to be much larger than that of the original state space. For example, this occurs for the HiPPO matrix [GGR22].

[5] As the inputs $u$ in Equation 5 are over $\mathbb{R}$ and we require the outputs to also be over $\mathbb{R}$ we explicitly cast the complex-valued kernel produced by Proposition 1 to $\mathbb{R}$ without affecting the soundness (details in §A.5).

The DSS layer can be implemented in just a few lines of code and our PyTorch implementation of $DSS_{SOFTMAX}$ layer is provided in §A.7 (Figure 6). The implementation of $DSS_{EXP}$ layer is even simpler and is omitted.

**Complexity**    For batch size $B$, sequence length $L$ and hidden size $H$, the DSS layer requires $O(NHL)$ time and space to compute the kernels, $O(BHL\log(L))$ time for the discrete convolution and $O(BH^2L)$ time for the output projection. For small batch size $B$, the time taken to compute the kernels becomes important whereas for large batches more compute is spent on the convolution and the linear projection. The kernel part of DSS layer has $2N + H + 2HN$ real-valued parameters.

## 3.3    Initialization of DSS layer

The performance of state spaces models is known to be highly sensitive to initialization [GGR22]. In line with the past work, we found that carefully initializing the parameters of the DSS layer is crucial to obtain state-of-the-art performance (§4).

The real and imaginary parts of each element of $W$ are initialized from $\mathcal{N}(0, 1)$. Each element of $\Delta_{\log}$ is initialized as $e^r$ where $r \sim \mathcal{U}(\log(.001), \log(.1))$. $\Lambda \in \mathbb{C}^N$ is initialized using eigenvalues of the normal part of normal plus low-rank form of HiPPO matrix [GGR22]. Concretely, $\Lambda_{\mathrm{re}}, \Lambda_{\mathrm{im}}$ are initialized such that the resulting $\Lambda$ is the vector of those $N$ eigenvalues of the following $2N \times 2N$ matrix which have a positive imaginary part.

$$
\begin{cases}
(2i + 1)^{1/2}(2j + 1)^{1/2}/2 & i < j \\
-1/2 & i = j \\
-(2i + 1)^{1/2}(2j + 1)^{1/2}/2 & i > j
\end{cases}
$$

Henceforth, we would refer to the above initialization of $\Lambda$ as *Skew-Hippo* initialization.[6] In all our experiments, we used the above initialization with $N = 64$. The initial learning rate of all DSS parameters was $10^{-3}$ and weight decay was not applied to them. Exceptions to these settings are noted in §A.3.

## 3.4    States of DSS via the Recurrent View

In §3, we argued that it can be slow to compute the output of state spaces via Equation 2 and instead leveraged Equation 5 for fast computation. But in situations such as autoregressive decoding during inference, it is more efficient to explicitly compute the states of a state space model similar to a linear RNN (Equation 2). We now show how to compute the states of the DSS models described in §3.1. Henceforth, we assume that we have already computed $\Lambda$ and $\Delta$.

**$DSS_{EXP}$**    As stated in Proposition 1(a), $DSS_{EXP}$ computes $\overline{K}_{\Delta,L}(\Lambda, (1)_{1 \leqslant i \leqslant N}, \tilde{w})$. For this state space and sample time $\Delta$, we use Equation 2 to obtain its discretization

$$
\overline{A} = \mathrm{diag}(e^{\lambda_1 \Delta}, \ldots, e^{\lambda_N \Delta}) \quad , \quad \overline{B} = \left(\lambda_i^{-1}(e^{\lambda_i \Delta} - 1)\right)_{1 \leqslant i \leqslant N}
$$

where $\mathrm{diag}$ creates a diagonal matrix of the scalars. We can now compute the states using the SSM recurrence $x_k = \overline{A}x_{k-1} + \overline{B}u_k$ (Equation 2). As $\overline{A}$ is diagonal, the $N$ coordinates do not interact and hence can be computed independently. Let us assume $x_{-1} = 0$ and say we have already computed $x_{k-1}$. Then, for the $i$'th coordinate independently compute

$$
x_{i,k} = e^{\lambda_i \Delta} x_{i,k-1} + \lambda_i^{-1}(e^{\lambda_i \Delta} - 1)u_k \ .
$$

Note that in $DSS_{EXP}$ , $\mathrm{Re}(\lambda_i) < 0$ and hence $|e^{\lambda_i \Delta}| = |e^{\mathrm{Re}(\lambda_i)\Delta}| < 1$. Intuitively, if $|\lambda_i|\Delta \approx 0$, we would have $x_{i,k} \approx x_{i,k-1}$ and be able to copy history over many timesteps. On the other hand if $\mathrm{Re}(\lambda_i)\Delta \ll 0$ then $x_{i,k} \approx -\lambda_i^{-1}u_k$ and hence the information from the previous timesteps would be forgotten similar to a "forget" gate in LSTMs.

---

   [6] Following our work, [GGGR22] formalized the connection between Skew-Hippo initialization and the original HiPPO matrices and provided a theoretical justification for its empirical properties.

**DSS$_{\text{SOFTMAX}}$**   As stated in Proposition 1(b), DSS$_{\text{SOFTMAX}}$ computes $\overline{K}_{\Delta,L}(\Lambda,\ ((e^{L\lambda_i\Delta} - 1)^{-1})_{1\leqslant i\leqslant N},\ w)$. For this state space and sample time $\Delta$, we obtain the discretization

$$\overline{A} = \mathrm{diag}(e^{\lambda_1\Delta}, \ldots, e^{\lambda_N\Delta}) \quad , \quad \overline{B} = \left(\frac{e^{\lambda_i\Delta} - 1}{\lambda_i(e^{\lambda_i\Delta L} - 1)}\right)_{1\leqslant i\leqslant N} .$$

For the $i$'th coordinate we can independently compute

$$x_{i,k} = e^{\lambda_i\Delta}x_{i,k-1} + \frac{u_k(e^{\lambda_i\Delta} - 1)}{\lambda_i(e^{\lambda_i\Delta L} - 1)} .$$

Let us drop the coordinate index $i$ for clarity to obtain

$$x_k = e^{\lambda\Delta}x_{k-1} + \frac{u_k(e^{\lambda\Delta} - 1)}{\lambda(e^{\lambda\Delta L} - 1)}$$

where $x_{k-1}$ is now a scalar. As the expression involves the term $e^{\lambda\Delta L}$, where $L$ can be large, directly computing such terms can result in numerical instability and we must avoid exponentiating scalars with a positive real part. We make two cases depending on the sign of $\mathrm{Re}(\lambda)$ and compute $x_k$ via an intermediate state $\widetilde{x}_k$ as follows. Let $p = I[\mathrm{Re}(\lambda) > 0] \in \{0,1\}$. Then,

$$\widetilde{x}_k = e^{\lambda\Delta(1-p)} \cdot \widetilde{x}_{k-1} + e^{-k\lambda\Delta p} \cdot u_k \quad , \quad x_k = \widetilde{x}_k \cdot \frac{e^{\lambda\Delta p(k-(L-1))}}{\lambda} \cdot \frac{e^{\lambda\Delta(1-2p)} - 1}{e^{\lambda\Delta(1-2p)L} - 1} .$$

The above equation can be parsed as follows. If $\mathrm{Re}(\lambda) \leqslant 0$ then we have

$$\widetilde{x}_k = e^{\lambda\Delta} \cdot \widetilde{x}_{k-1} + u_k \quad , \quad x_k = \widetilde{x}_k \cdot \frac{(e^{\lambda\Delta} - 1)}{\lambda(e^{\lambda\Delta L} - 1)}$$

and hence if $\mathrm{Re}(\lambda)\Delta \ll 0$ then $\widetilde{x}_k \approx u_k$ and $x_k \approx u_k/\lambda$. In this case, information from the previous timesteps would be ignored and the information used would be local. On the other hand if $\mathrm{Re}(\lambda) > 0$ then we would have

$$\widetilde{x}_k = \widetilde{x}_{k-1} + e^{-k\lambda\Delta} \cdot u_k \quad , \quad x_k = \widetilde{x}_k \cdot \frac{e^{\lambda\Delta(k-(L-1))}}{\lambda} \cdot \frac{e^{-\lambda\Delta} - 1}{e^{-\lambda\Delta L} - 1}$$

and hence if $\mathrm{Re}(\lambda)\Delta \gg 0$ then $\widetilde{x}_0 \approx u_0$ and $\widetilde{x}_k \approx \widetilde{x}_{k-1} \approx u_0$, $x_{k<L-1} \approx 0$ and $x_{L-1} \approx u_0/\lambda$. Hence, the model would be able to copy information even from extremely distant positions, allowing it to capture long-range dependencies.

## 4   Experiments

We evaluate the performance of DSS on sequence-level classification tasks over text, images and audio. Overall, we find its performance is comparable to S4.

**Long Range Arena (LRA)**   LRA [TDA$^+$21] is a standard benchmark for assessing the ability of models to process long sequences. LRA contains 6 tasks with diverse input lengths $1K$-$16K$, encompassing modalities such as text and images. Several Transformer variants have been benchmarked on LRA but all have underperformed due to factors such as their high compute-memory requirements, implicit locality bias and inability to capture long-range dependencies.

Table 1 compares DSS against S4, the Transformer variants reported in [TDA$^+$21], as well as follow-up work. State space models (S4, DSS) shown in Table 1 are left-to-right *unidirectional* whereas other models could be bidirectional.

Despite its simplicity, DSS delivers state-of-the-art performance on LRA. Its performance is comparable to that of S4, with a modest improvement in test accuracy averaged across the 6 tasks (81.88 vs 80.21).[7] In addition, DSS maintains a comfortable 20 point lead over the best performing Transformer variant (81.88 vs 61.41).

---

[7]   The large gap between S4 and DSS on TEXT is due to the use of a larger learning rate for $\Delta_{\log}$ in DSS. For our S4 runs, we decided to use the official hyperparameters as provided by [GGR22].

| MODEL | LISTOPS | TEXT | RETRIEVAL | IMAGE | PATHFINDER | PATH-X | AVG |
|---|---|---|---|---|---|---|---|
| (input length) | 2000 | 2048 | 4000 | 1024 | 1024 | 16384 | |
| Transformer | 36.37 | 64.27 | 57.46 | 42.44 | 71.40 | ✗ | 53.66 |
| Reformer | 37.27 | 56.10 | 53.40 | 38.07 | 68.50 | ✗ | 50.56 |
| BigBird | 36.05 | 64.02 | 59.29 | 40.83 | 74.87 | ✗ | 54.17 |
| Linear Trans. | 16.13 | 65.90 | 53.09 | 42.34 | 75.30 | ✗ | 50.46 |
| Performer | 18.01 | 65.40 | 53.82 | 42.77 | 77.05 | ✗ | 51.18 |
| FNet | 35.33 | 65.11 | 59.61 | 38.67 | 77.80 | ✗ | 54.42 |
| Nyströmformer | 37.15 | 65.52 | 79.56 | 41.58 | 70.94 | ✗ | 57.46 |
| Luna-256 | 37.25 | 64.57 | 79.29 | 47.38 | 77.72 | ✗ | 59.37 |
| H-Transformer-1D | 49.53 | 78.69 | 63.99 | 46.05 | 68.78 | ✗ | 61.41 |
| S4 (as in [GGR22]) | 58.35 | 76.02 | 87.09 | **87.26** | 86.05 | **88.10** | 80.48 |
| S4 (**our run**) | 57.6 | 75.4 | 87.6 | 86.5 | **86.2** | 88.0 | 80.21 |
| **DSS**$_{\text{SOFTMAX}}$ (**ours**) | **60.6** | **84.8** | **87.8** | 85.7 | 84.6 | 87.8 | **81.88** |
| **DSS**$_{\text{EXP}}$ (**ours**) | 59.7 | 84.6 | 87.6 | 84.9 | 84.7 | 85.6 | 81.18 |
| **DSS**$_{\text{EXP-NO-SCALE}}$ (**ours**) | 59.3 | 82.4 | 86.0 | 81.2 | 81.3 | ✗ | 65.03 |

Table 1: (**Long Range Arena**) Accuracy on the full suite of LRA tasks. (*Top*) Transformer variants reported in LRA, (*Middle*) other long-range models reported in the literature, (*Bottom*) state space models.

| MODEL | MFCC | RAW | | MODEL | MFCC | RAW |
|---|---|---|---|---|---|---|
| Transformer | 90.75 | ✗ | | WaveGAN-D | ✗ | 96.25 |
| Performer | 80.85 | 30.77 | | S4 (as in [GGR22]) | 93.96 | **98.32** |
| ODE-RNN | 65.9 | ✗ | | S4 (**our run**) | | 98.1 |
| NRDE | 89.8 | 16.49 | | **DSS**$_{\text{SOFTMAX}}$ (**ours**) | | 97.7 |
| ExpRNN | 82.13 | 11.6 | | **DSS**$_{\text{EXP}}$ (**ours**) | | 98.2 |
| LipschitzRNN | 88.38 | ✗ | | **DSS**$_{\text{EXP-NO-SCALE}}$ (**ours**) | | 97.7 |
| CKConv | **95.3** | 71.66 | | | | |

Table 2: (**Speech Commands (SC)**) Transformer, CTM, RNN, CNN, and SSM models. (*MFCC*) Standard pre-processed MFCC features (length-161). (*Raw*) Unprocessed signals (length-16000). ✗ denotes not applicable or computationally infeasible on a single GPU. All results except ours are reproduced from [GGR22].

Interestingly, despite being less expressive than DSS$_{\text{SOFTMAX}}$, DSS$_{\text{EXP}}$ also reports an impressive performance which makes it specially appealing given its simplicity during training and inference. We investigate an even simpler version of DSS$_{\text{EXP}}$, denoted as DSS$_{\text{EXP-NO-SCALE}}$, which is same as DSS$_{\text{EXP}}$ except that we omit the term $\Lambda^{-1}(e^{\Lambda\Delta} - I)$ in the expression of the kernel shown in Proposition 1(a) and instead compute it as $\widetilde{w} \cdot \text{elementwise-exp}(P)$. As shown in Table 1, the performance of DSS$_{\text{EXP-NO-SCALE}}$ is generally inferior to that of DSS$_{\text{EXP}}$ with the model failing on the challenging PATH-X task which requires the model to capture extremely long-range interactions.

**Raw Speech Classification**   Audio is typically digitized using a high sampling rate resulting in very long sequences. This provides an interesting domain for investigating the abilities of long-range models. We evaluate the performance of DSS on the Speech Commands (SC) dataset [War18], consisting of raw audio samples of length 16000, modeled as a 10-way classification task. As shown in Table 2, the performance of all DSS variants is comparable to that of S4 (98.2 vs 98.1).

In all experiments and ablations, S4 and DSS use identical model hyperparameters such as hidden size, number of layers, etc. Our experimental setup was built on top of the training framework provided by the S4 authors and for our S4 runs we followed their official instructions.[8] Details about model initialization, and hyperparameters are provided in §A.3.

## 4.1   Analyzing the Performance of DSS

While the experimental results presented above are encouraging, and clearly demonstrate the effectiveness of DSS at modeling long-range dependencies, it is not clear what exactly are the main factors

---

[8] https://github.com/HazyResearch/state-spaces

| MODEL | LISTOPS | TEXT | IMAGE | PATHFINDER | PATH-X | SC |
|---|---|---|---|---|---|---|
| Non-State-Space best | 49.5 | 78.7 | 47.4 | 77.8 | ✗ | 96.5 |
| DSS$_{\text{SOFTMAX}}$ | **60.6** | **84.8** | **85.7** | **84.6** | **87.8** | **97.7** |
| DSS$_{\text{SOFTMAX}}$ (**random init**) | 57.6 | 80.54 | 72.13 | 77.5 | ✗ | 96.8 |
| DSS$_{\text{SOFTMAX}}$ (**kernel length** 128) | 51.6 | 75.41 | 83.8 | 65.1 | ✗ | 96.7 |
| DSS$_{\text{SOFTMAX}}$ ($\text{argmax}_{k<L}|K_k|$-95-percentile) | 89 | 124 | 66 | 87 | 1269 | 81 |

Table 3: *(Top)* Ablation analysis of DSS$_{\text{SOFTMAX}}$ . "Non-State-Space best" is the best performance among models in top and middle sections of Table 1. *(Bottom)* "$\text{argmax}_{k<L}|K_k|$-95-percentile" is the 95th percentile of $\{\text{argmax}_{0\leqslant k<L}|K_k| : K$ is a kernel in DSS$_{\text{SOFTMAX}}$ $\}$ and is explained in §4.2.

contributing to its performance. To investigate this further, we performed an ablation analysis aimed at answering the following questions:

- How significant is the Skew-Hippo initialization (§3.3) to the model performance? Would initializing $\Lambda$ randomly work just as well?

- Is the main source of superior performance of state space models (S4, DSS), compared to previous models, their ability to model long-range dependencies? Would restricting DSS to only model local interactions hurt its performance on the above tasks?

**Random Initialization**   To answer the first question, we repeated the experiments after initializing each element of $\Lambda_{\text{re}}$ and $\Lambda_{\text{im}}$ in DSS$_{\text{SOFTMAX}}$ by randomly sampling from $\mathcal{N}(0,1)$.

**Truncated Kernels**   To answer the second question, instead of constructing a kernel of length equal to the length $L$ of the input, we restricted the length of the kernel constructed in DSS$_{\text{SOFTMAX}}$ (Algorithm 1) to 128, significantly shorter than the length of the input. To understand the implication of this restriction recall Equation 5 which states that $y_k = \sum_{j=0}^{k} \overline{K}_j \cdot u_{k-j}$.

For a given context size $c = 128$, restricting $\overline{K}_{\geqslant c} = 0$ would imply

$$y_k = \sum_{j=0}^{\min(k,c-1)} \overline{K}_j \cdot u_{k-j}$$

and hence the output $y_k$ at position $k$ would only depend on $u_k, \ldots, u_{k-c+1}$. This would restrict each DSS$_{\text{SOFTMAX}}$ layer to only model local interactions and the model would require several layers to have a broader receptive field and capture long-range interactions.

As shown in Table 3, randomly initializing the $\Lambda$ parameters of DSS leads to significant performance degradation on the majority of tasks, with the model failing to perform on PATH-X. This is inline with the findings of [GGR22] who also reported the initialization of S4 to be critical to its performance. Interestingly, despite this performance reduction, DSS manages to outperform all non state-space-based models on every task.

Truncating the length of the kernel also leads to a significant reduction in performance across most tasks (Table 3), suggesting that the superior performance of state-space models on these tasks can indeed be partly attributed to their ability to capture long-range dependencies. Moreover, on some tasks such as LISTOPS and IMAGE, using a truncated kernel still manages to outperform all Transformer variants, which is surprising as Transformer layers are known to be effective at capturing interactions at such short ranges.

## 4.2   Analysis of Learned DSS Parameters

To further explore the inner workings of DSS, we visually inspected the trained parameters and kernels of DSS$_{\text{SOFTMAX}}$ .

The kernels of all layers of the trained DSS$_{\text{SOFTMAX}}$ are shown in Figure 2 and reveal a stark contrast between the tasks. On the tasks IMAGE and SC, for almost all kernels, the absolute values of the first 128 positions are significantly higher than the later positions indicating that these kernels are mostly local. On the other hand for PATH-X, for a significant proportion of kernels the opposite is true, indicating that these kernels are modeling long-range interactions.

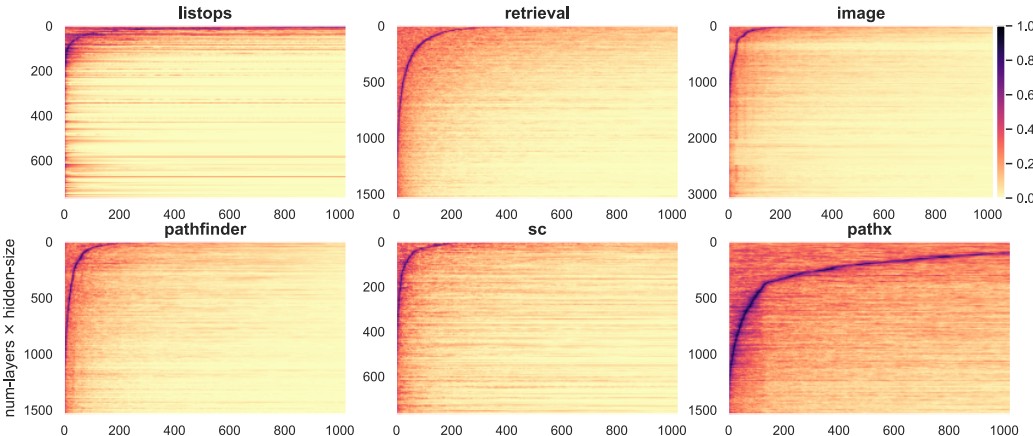

Figure 2: Kernels of trained $\text{DSS}_{\text{SOFTMAX}}$. Each row of pixels corresponds to one of the (number-of-layers $\times H$) kernels. For a row (kernel) $K \in \mathbb{R}^L$ the element $K_k \in \mathbb{R}$ is shown as $|K_k|/||K||_\infty$ to enable visualization. To enable comparison across tasks, only the first 1024 positions are shown.

This partially explains why the performance of $\text{DSS}_{\text{SOFTMAX}}$ on IMAGE and SC does not decrease significantly even after limiting the kernel length to 128, whereas PATHFINDER and PATH-X report significant degradation (Table 3). The last row of Table 3 shows the 95th percentile of the set $\{\arg\max_{0 \leqslant k < L} |K_k| : K$ is a kernel in $\text{DSS}_{\text{SOFTMAX}}\}$, i.e., the set of positions over all kernels, at which for a given kernel its absolute value is maximized. As expected, most kernels of IMAGE and SC are local whereas for PATH-X they are mostly long range.

The values of the real and imaginary parts of $\Lambda$ are shown in Figure 3 (and §A.6, Figure 5). We observe that, although all elements of $\Lambda_{\text{re}}$ are initialized as $-0.5$, their values can change significantly during training and can become positive (see plots for LISTOPS, SC). It can also be observed that a $\lambda$ with a more negative $\text{Re}(\lambda)$ generally tends to have a larger $\text{Im}(\lambda)$, which interestingly is a property not satisfied by Skew-Hippo initialization.

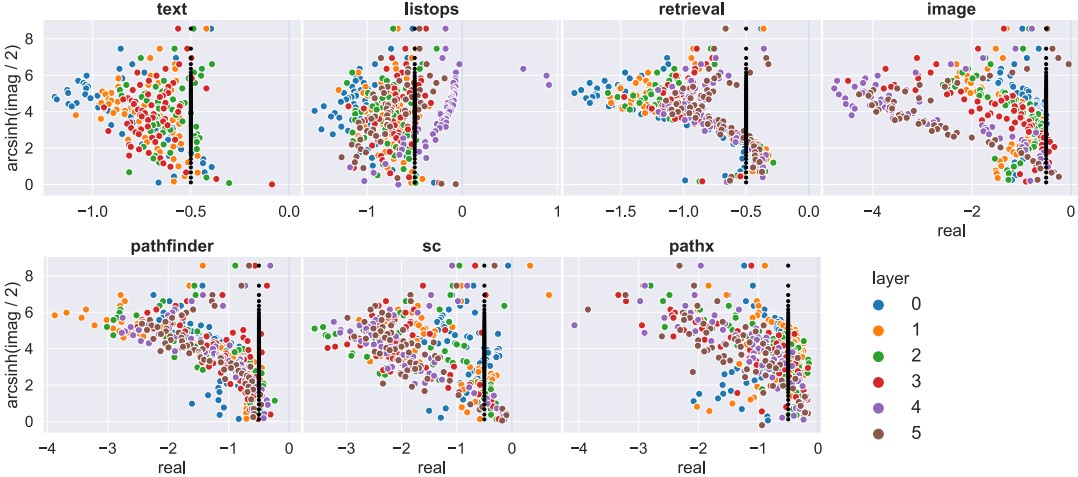

Figure 3: Values of $\Lambda$ in trained $\text{DSS}_{\text{SOFTMAX}}$ for tasks described in §4. For $\lambda = x + iy$, we show $y$ on log-scale for better visualization by plotting $\lambda$ as $(x, \text{arcsinh}(y/2)) = (x, \log(y/2 + \sqrt{(y/2)^2 + 1}))$. Black dots correspond to Skew-Hippo initialization (§3.3).

The values of the trained $\Delta_{\text{log}}$ corresponding to the real and imaginary parts of $\Lambda$ are shown in Figure 4 and, in this case as well, change significantly during training. The values of $\Delta_{\text{log}}$ on short-range tasks such as IMAGE and SC generally tend to be larger compared to long-range tasks such as PATHFINDER and PATH-X, inline with the intuition that, when $\text{Re}(\lambda) < 0$, smaller values of $\Delta$ correspond to modeling long-range dependence (§3.4).

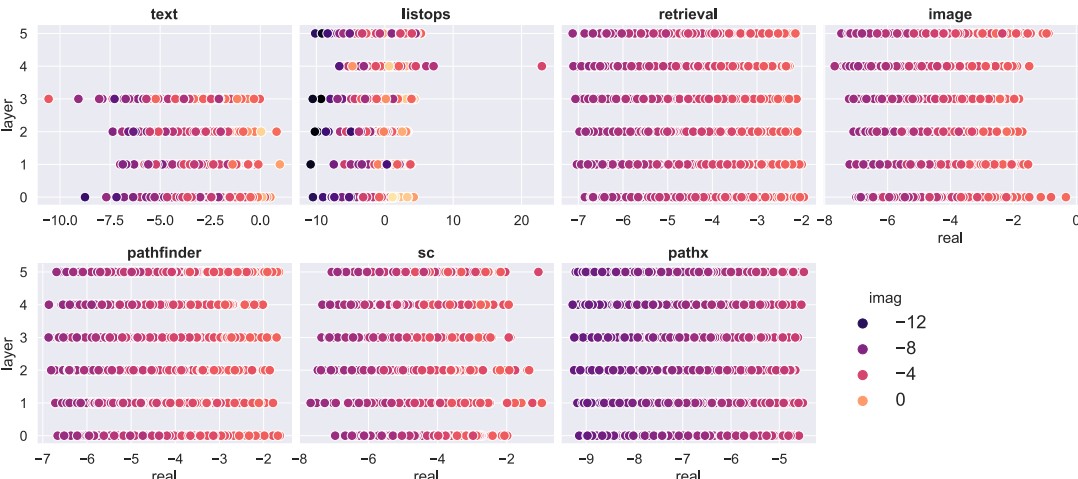

Figure 4: Trained $\Delta_{\log}$ in $\text{DSS}_{\text{SOFTMAX}}$ for tasks described in §4. As noted in §A.3 we used separate $\Delta_{\log,\text{re}}$, $\Delta_{\log,\text{im}} \in \mathbb{R}$ parameters to respectively scale the real and imaginary parts of $\Lambda$.

We note that the plot for LISTOPS reveals an outlier with a value of 22 which after exponentiation in Algorithm 1 would result in an extreme large $\Delta$. This can potentially lead to training instabilities and we plan to address this issue in future work.

**Limitations and future work** In this work, we evaluated DSS on sequence-level classification tasks. In future work, we plan to include token-level generation tasks such as language modeling, forecasting, etc. Another important future direction is to pretrain DSS-based models on large amounts of raw data. Lastly, we found that the initialization and learning rates of DSS parameters $\Lambda_{\text{re}}$, $\Lambda_{\text{im}}$, $\Delta_{\log}$ (§3.3) play an important role in the performance and convergence of the model. Informally, for tasks such as PATH-X that require very long-range interactions, a smaller initialization of $\Delta_{\log}$ was beneficial. An in-depth analysis of this phenomenon could be helpful and remains for future work.

## Acknowledgments and Disclosure of Funding

We thank Ramon Fernandez Astudillo for carefully reviewing the preliminary draft and suggesting several helpful edits. We thank Omer Levy, Achille Fokoue and Luis Lastras for their support. Our experiments were conducted on IBM's Cognitive Computing Cluster, with additional resources from Tel Aviv University. This research was supported by (1) IBM AI Residency program and (2) Defense Advanced Research Projects Agency (DARPA) through Cooperative Agreement D20AC00004 awarded by the U.S. Department of the Interior (DOI), Interior Business Center, and (3) the European Research Council (ERC) under the European Union Horizons 2020 research and innovation programme (grant ERC DELPHI 802800).

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
