# A Supplemental Material

## A.1 Diagonal State Spaces

We restate Proposition 1 for convenience.

**Proposition.** *Let $K \in \mathbb{C}^{1 \times L}$ be the kernel of length $L$ of a given state space $(A, B, C)$ and sample time $\Delta > 0$, where $A \in \mathbb{C}^{N \times N}$ is diagonalizable over $\mathbb{C}$ with eigenvalues $\lambda_1, \ldots, \lambda_N$ and $\forall i, \lambda_i \neq 0$ and $e^{L\lambda_i \Delta} \neq 1$. Let $P \in \mathbb{C}^{N \times L}$ be $P_{i,k} = \lambda_i k\Delta$ and $\Lambda$ be the diagonal matrix with $\lambda_1, \ldots, \lambda_N$. Then there exist $\widetilde{w}, w \in \mathbb{C}^{1 \times N}$ such that*

(a) $K = \overline{K}_{\Delta, L}(\Lambda, (1)_{1 \leqslant i \leqslant N}, \widetilde{w}) = \widetilde{w} \cdot \Lambda^{-1}(e^{\Lambda \Delta} - I) \cdot \text{elementwise-exp}(P),$

(b) $K = \overline{K}_{\Delta, L}(\Lambda, ((e^{L\lambda_i \Delta} - 1)^{-1})_{1 \leqslant i \leqslant N}, w) = w \cdot \Lambda^{-1} \cdot \text{row-softmax}(P).$

*Proof.* Let $A$ be diagonalizable over $\mathbb{C}$ as $A = V\Lambda V^{-1}$ with eigenvalues $\lambda_1, \ldots, \lambda_N \in \mathbb{C}$. From Equation 4 we have

$$K = (Ce^{A \cdot k\Delta}(e^{A\Delta} - I)A^{-1}B)_{0 \leqslant k < L},$$

where

$$K_k = Ce^{A \cdot k\Delta}(e^{A\Delta} - I)A^{-1}B = (CV)e^{\Lambda k\Delta}(e^{\Lambda \Delta} - I)\Lambda^{-1}(V^{-1}B).$$

For $CV \in \mathbb{C}^{1 \times N}$ and $V^{-1}B \in \mathbb{C}^{N \times 1}$ let $(CV)^{\top} * (V^{-1}B) = \widetilde{w} \in \mathbb{C}^N$ be the element-wise product of $CV$ and $V^{-1}B$. Then,

$$K_k = \sum_{i=1}^{N} \frac{e^{\lambda_i k\Delta}(e^{\lambda_i \Delta} - 1)}{\lambda_i} \cdot \widetilde{w}_i \tag{7}$$

$$= \sum_{i=1}^{N} \frac{e^{\lambda_i k\Delta}(e^{\lambda_i \Delta} - 1)}{\lambda_i(e^{L\lambda_i \Delta} - 1)} \cdot ((e^{L\lambda_i \Delta} - 1)\widetilde{w}_i) \tag{8}$$

$$= \sum_{i=1}^{N} (\widetilde{w}_i \cdot (e^{L\lambda_i \Delta} - 1)) \cdot \frac{1}{\lambda_i} \cdot \frac{e^{\lambda_i k\Delta}}{(\sum_{r=0}^{L-1} e^{r\lambda_i \Delta})} \tag{9}$$

where the last equality follows from $(z^L - 1) = (z - 1)(z^0 + \ldots + z^{L-1})$ and using $z^L \neq 1$.

Let $P \in \mathbb{C}^{N \times L}$ be the matrix $P_{i,k} = \lambda_i \cdot k\Delta$ and let $E = \text{elementwise-exp}(P)$. It is easy to verify that Equation 7 can be re-written as a vector-matrix product as

$$K = \widetilde{w} \cdot \Lambda^{-1}(e^{\Lambda \Delta} - I) \cdot E.$$

Similarly, for the state space $(\Lambda, (1)_{1 \leqslant i \leqslant N}, \widetilde{w}))$ and sample time $\Delta$ its kernel $\widetilde{K}$ can be obtained from Equation 4 as

$$\widetilde{K}_k = \widetilde{w} \cdot e^{\Lambda \cdot k\Delta}(e^{\Lambda \Delta} - I)\Lambda^{-1} \cdot [1, \ldots, 1]_{N \times 1}$$

$$= \sum_{i=1}^{N} \widetilde{w}_i \cdot \frac{e^{\lambda_i k\Delta}(e^{\lambda_i \Delta} - 1)}{\lambda_i}$$

which is also the expression for $K_k$ (Equation 7). This proves part (a) and we now consider part (b). Let $w \in \mathbb{C}^N$ be defined as

$$w_i = \widetilde{w}_i \cdot (e^{L\lambda_i \Delta} - 1).$$

Then from Equation 9,

$$K_k = \sum_{i=1}^{N} w \cdot \frac{1}{\lambda_i} \cdot \frac{e^{\lambda_i k\Delta}}{(\sum_{r=0}^{L-1} e^{r\lambda_i \Delta})}. \tag{10}$$

Let $S = \text{row-softmax}(P)$ denote the matrix obtained after applying $\text{softmax}$ on the rows of $P$, i.e.

$$S_{i,k} = \frac{e^{\lambda_i k\Delta}}{\sum_{r=0}^{L-1} e^{r\lambda_i \Delta}}.$$

It is easy to verify that Equation 10 can be expressed as a vector-matrix product

$$K = w \cdot \Lambda^{-1} \cdot S \,.$$

Similarly, for the state space $(\Lambda,\ ((e^{L\lambda_i\Delta} - 1)^{-1})_{1 \leqslant i \leqslant N},\ w))$ and sample time $\Delta$ its kernel $\widehat{K}$ can be obtained from Equation 4 as

$$
\begin{aligned}
\widehat{K}_k &= w \cdot e^{\Lambda \cdot k\Delta}(e^{\Lambda\Delta} - I)\Lambda^{-1} \cdot [\dots, (e^{L\lambda_i\Delta} - 1)^{-1}, \dots]_{N\times 1} \\
&= \sum_{i=1}^{N} w_i \cdot \frac{e^{\lambda_i k\Delta}(e^{\lambda_i\Delta} - 1)}{\lambda_i(e^{L\lambda_i\Delta} - 1)} \\
&= \sum_{i=1}^{N} \widetilde{w}_i \cdot \frac{e^{\lambda_i k\Delta}(e^{\lambda_i\Delta} - 1)}{\lambda_i}
\end{aligned}
$$

which is also the expression for $K_k$ (Equation 7).  $\qquad\square$

## A.2   Numerically Stable softmax

As noted in §3.1, softmax can have singularities over $\mathbb{C}$. To address this issue, we use a simple correction to make it well-defined over the entire domain:

- softmax : Given $(x_0, \dots, x_{L-1}) = x \in \mathbb{C}^L$, let $\mathrm{softmax}(x) \in \mathbb{C}^L$ be defined as $(\mathrm{softmax}(x))_k = e^{x_k}(e^{x_0} + \dots + e^{x_{L-1}})^{-1}$. Note that for any $c \in \mathbb{C}$, $\mathrm{softmax}(x_0, \dots, x_{L-1}) = \mathrm{softmax}(x_0 - c, \dots, x_{L-1} - c)$. Unlike over $\mathbb{R}$, softmax can have singularities over $\mathbb{C}$ as sum of exponentials can vanish. E.g. $e^0 + e^{i\pi} = 0$ and hence $\mathrm{softmax}(0, i\pi)$ is not defined.

- max : Given $(x_0, \dots, x_{L-1}) = x \in \mathbb{C}^L$, let $\max(x)$ be the $x_i$ with the maximum real part, i.e. $x_{\mathrm{argmax}_i \mathrm{Re}(x_i)}$.

- reciprocal$_\epsilon$ : Given $x \in \mathbb{C}$ and $\epsilon \in \mathbb{R}_{>0}$, let $\mathrm{reciprocal}_\epsilon(x) = \frac{\overline{x}}{x \cdot \overline{x} + \epsilon}$ where $\overline{x}$ is the complex conjugate of $x$. The denominator is always in $\mathbb{R}_{\geqslant \epsilon}$ and $|\mathrm{reciprocal}_\epsilon| \leqslant (2\sqrt{\epsilon})^{-1}$.

- softmax$_\epsilon$ : Given $(x_0, \dots, x_{L-1}) = x \in \mathbb{C}^L$ let $m = \max(x)$ and $\widetilde{x}_i = x_i - m$. Note that $|e^{\widetilde{x}_i}| \leqslant 1$. Given $\epsilon \in \mathbb{R}_{>0}$, let $\mathrm{softmax}_\epsilon(x) \in \mathbb{C}^L$ be

$$(\mathrm{softmax}_\epsilon(x))_k = e^{\widetilde{x}_k} \cdot \mathrm{reciprocal}_\epsilon\left(\sum_{r=0}^{L-1} e^{\widetilde{x}_r}\right).$$

  softmax$_\epsilon$ is always bounded and differentiable.

In our implementation, we use softmax$_\epsilon$ with $\epsilon = 10^{-7}$.

**SSM Softmax**   In our current implementation of $\mathrm{softmax}(x)$ (Figure 6), we exploit the specific structure of $x$ that arises in Algorithm 1. We now describe an alternate method based on FFT which also uses this specific structure and might lead to a faster implementation in the future.

**Claim 1** (SSM Softmax). *Given $c \in \mathbb{C}$, let $p = I[\mathrm{Re}(c) > 0]$, $n = 1 - p$, $e = \exp(c \cdot (n - p))$ and $r = (n - pe)/(p - ne)$. Let $\omega = \exp(-2\pi i/L)$ where $i = \sqrt{-1}$. Then,*

$$
\begin{aligned}
\mathrm{softmax}(c \cdot 0, \dots, c \cdot (L-1)) &= \mathrm{inverseFFT}\left(\frac{1-e}{n - pe + (p - ne)\omega^k}\right)_{0 \leqslant k < L} \\
&= \mathrm{inverseFFT}\left(\frac{r+1}{r + \omega^k}\right)_{0 \leqslant k < L}.
\end{aligned}
$$

*Proof.* There are 2 cases depending on sign of $\mathrm{Re}(c)$.

**Case 1** ($p = 0, n = 1$): In this case we have $e = \exp(c)$. For the map

$$F(z) = \frac{1-e}{1 - e^L} \sum_{k=0}^{L-1} (ez)^k = \frac{(1-e)(1 - (ez)^L)}{(1 - e^L)(1 - ez)}$$

we get the coefficients of $F(z)$ as

$$\text{invFFT}(F(\omega^k)_{0\leqslant k<L}) \quad = \quad \left(\frac{(1-e)}{(1-e^L)}e^k\right)_{0\leqslant k<L} \quad = \quad \text{softmax}(c\cdot 0,\ldots,c\cdot(L-1)).$$

We have,

$$F(\omega^k) \quad = \quad \frac{(1-e)(1-(e\omega^k)^L)}{(1-e^L)(1-e\cdot\omega^k)} \quad = \quad \frac{1-e}{1-e\cdot\omega^k}$$

where last equality follows from $\omega^L = 1$.

**Case 2** ($p = 1, n = 0$): In this case we have $e = \exp(-c)$. For the map

$$F(z) \quad = \quad \frac{1-e}{1-e^L}\sum_{k=0}^{L-1}e^k z^{L-1-k} \quad = \quad \frac{(1-e)z^{L-1}}{1-e^L}\sum_{k=0}^{L-1}\left(\frac{e}{z}\right)^k$$

$$= \quad \frac{(1-e)z^{L-1}}{1-e^L}\frac{1-\left(\frac{e}{z}\right)^L}{1-\frac{e}{z}} \quad = \quad \frac{(1-e)}{(1-e^L)}\frac{(z^L-e^L)}{(z-e)}$$

we get the coefficients of $F(z)$ as

$$\text{invFFT}(F(\omega^k)_{0\leqslant k<L}) \quad = \quad \left(\frac{(1-e)}{1-e^L}e^{L-1-k}\right)_k$$

$$= \quad \left(\frac{(e^{-1}-1)}{(e^{-1})^L-1}(e^{-1})^k\right)_{0\leqslant k<L} \quad = \quad \text{softmax}(c\cdot 0,\ldots,c\cdot(L-1))$$

as $e^{-1} = \exp(c)$. Moreover, we have

$$F(\omega^k) \quad = \quad \frac{(1-e)(\omega^{k\cdot L}-e^L)}{(1-e^L)(\omega^k-e)} \quad = \quad \frac{1-e}{-e+\omega^k}$$

where last equality follows from $\omega^L = 1$.

Finally, the second equality of the main Claim follows from $1 - e = n - pe + p - ne$.  $\square$

The computation of $\text{softmax}$ in Claim 1 is numerically stable and we always exponentiate scalars with a negative real part. The computed function has singularities at $c \in \{-2\pi ik/L, 0 \leqslant k < L\}$.

### A.3 Experimental Setup

We now describe the training details for DSS and S4 on LRA and Speech Commands (§4).

*Sequence Classification Head:* Both LRA and Speech Commands are sequence classification tasks. The final layer of the DSS stack outputs a sequence which is aggregated into a single vector via mean pooling along the length dimension. Exceptions to this were TEXT and PATHFINDER tasks where the rightmost token was used as the aggregate.

We used a separate $\Delta_{\log,\text{re}}, \Delta_{\log,\text{im}} \in \mathbb{R}$ parameters to respectively scale the real and imaginary parts of $\Lambda$. I.e. for a given $\Lambda \in \mathbb{C}^N$, we computed $\Delta * \Lambda$ as $\exp(\Delta_{\log,\text{re}})\Lambda_{\text{re}} + i\cdot\exp(\Delta_{\log,\text{im}})\Lambda_{\text{im}}$.

For all datasets, we used AdamW optimizer with a constant learning rate schedule with decay on validation plateau. However, for the DSS parameters (§3.2) initial learning rate was $10^{-3}$ and weight decay was not used, with a few exceptions noted below.

We used hyperparameters such as model sizes, number of update steps, etc as recommended by the S4 authors on their official repository and are listed in Table 4. We made the following exceptions for DSS trainings:

- LISTOPS: learning rate of $\Delta_{\log}$ was 0.02 instead of $10^{-3}$.
- TEXT: learning rate of $\Delta_{\log}$ 0.02 instead of $10^{-3}$.
- IMAGE: we used seed 0 and trained for 200 epochs instead of 100.

|  | Depth | Features $H$ | Norm | Pre-norm | Dropout | LR | Batch Size | Epochs | WD | Patience |
|---|---|---|---|---|---|---|---|---|---|---|
| **ListOps** | 6 | 128 | BN | False | 0 | 0.01 | 50 | 50 | 0.01 | 5 |
| **Text** | 4 | 128 | BN | True | 0 | 0.01 | 50 | 40 | 0 | 10 |
| **Retrieval** | 6 | 256 | BN | True | 0 | 0.002 | 64 | 25 | 0 | 20 |
| **Image** | 6 | 512 | LN | False | 0.2 | 0.004 | 50 | 200 | 0.01 | 10 |
| **Pathfinder** | 6 | 256 | BN | True | 0.1 | 0.004 | 100 | 200 | 0 | 10 |
| **Path-X** | 6 | 256 | BN | True | 0.0 | 0.0005 | 32 | 100 | 0 | 40 |
| **Speech Commands (Raw)** | 6 | 128 | BN | True | 0.1 | 0.01 | 20 | 200 | 0 | 20 |

Table 4: Hyperparameters for the S4 and DSS models. Exceptions for DSS are detailed in §A.3. (Top) LRA and (Bottom) Speech Commands. LR is initial learning rate and WD is weight decay. BN and LN refer to Batch Normalization and Layer Normalization.

- PATHFINDER: we used Patience $= 13$.
- PATH-X: we used batch size 16 and trained for 35 epochs. $\Delta_{\log}$ was initialized as $e^r$ where $r \sim \mathcal{U}(\log(.0001), \log(.01))$ and its learning rate was $10^{-4}$. This was beneficial in early convergence of the model.

For our experiments, the test accuracy that we report in §4 was measured at the checkpoint with the highest validation accuracy.

All our experiments were conducted on a single A100 GPU (40GiB).

## A.4 Benchmarking Running Times of DSS and S4

We compared the running time of DSS with that of the official S4 implementation on a single NVIDIA 3090 GPU. Implementations of both S4 and DSS versions below utilize the PyKeOps library for memory efficiency [CFG+21].

- Running times on the PATH-X task with input length $L = 16384$ are summarized in Table 5. For large batches the time taken to perform the FFT-based convolution and the feedforward part dominates and hence the speedups of DSS over S4 are less pronounced. For smaller $B$, the time taken to compute the kernel becomes significant and DSS provides upto $1.8\times$ speedup over S4.
- We also isolated and benchmarked the time taken to solely compute the S4 kernel vs the DSS kernel (including the time to perform a backward pass on the sum of all kernel entries). As summarized in Table 6, for large $L$, DSS kernel is more than $2\times$ faster than S4 kernel.

| $B$ | 1 | 16 |
|---|---|---|
| S4 | 133 | 470 |
| $\text{DSS}_{\text{EXP}}$ | 72 | 400 |
| $\text{DSS}_{\text{SOFTMAX}}$ | 75 | 410 |

| $L$ | 4096 | 16384 | 65536 |
|---|---|---|---|
| S4 | 4.3 | 10.8 | 39.7 |
| $\text{DSS}_{\text{EXP}}$ | 4.2 | 7.0 | 18.1 |
| $\text{DSS}_{\text{SOFTMAX}}$ | 4.3 | 6.9 | 17.8 |

Table 5: Time (msec) taken for a single gradient update on PATH-X using hyperparameters in Table 4 and batch size $B$.

Table 6: Time (msec) taken to compute the kernel of length $L$ for $N = 64$, $H = 256$.

## A.5 Additional Remarks

**Casting the kernel to $\mathbb{R}$**   In this work, we require the inputs $u$ and outputs $y$ to always be over $\mathbb{R}$, i.e., we assume Equation 5 to be with an explicit casting operation $y_k = \text{Re}(\sum_{j=0}^{k} \overline{K}_j \cdot u_{k-j})$. For $u$ over $\mathbb{R}$, this further implies $y_k = \sum_{j=0}^{k} \text{Re}(\overline{K}_j) \cdot u_{k-j}$. Hence, we can explicitly cast the complex-valued kernel produced by Proposition 1 to $\mathbb{R}$ during training (Algorithm 1, final step) and cast the outputs over $\mathbb{C}$ to $\mathbb{R}$ during inference in §3.4.

## A.6 Learned Parameters of DSSSOFTMAX

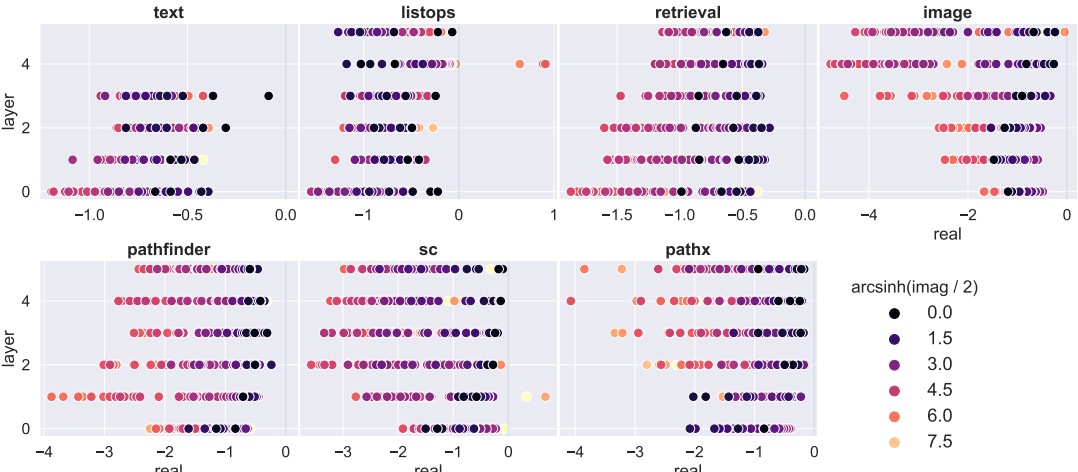

Figure 5: Trained $\Lambda$ in DSSSOFTMAX for tasks described in §4.

## A.7 Implementation of DSSSOFTMAX

```python
def reciprocal(x, epsilon=1e-7):
    x_conj = x.conj()          # conjugate
    return x_conj / (x*x_conj + epsilon)

def dss_kernel(L):
    # L: kernel length
    # Lambda: [N 2],   log_dt: [H],    W: [H N 2]   (floats)
    Lambda, log_dt, W = get_layer_parameters()
    # complex parameter stored as 2 floats denoting real,
    # imaginary parts as ADAM moments are non-linear

    # convert reals to complex
    Lambda, W = map(torch.view_as_complex, (Lambda, W))     # [N], [H N]
    dt_Lambda = log_dt.exp().unsqueeze(-1) * Lambda          # [H L]
    pos = torch.arange(L, device=W.device)                   # [L]
    P = dt_Lambda.unsqueeze(-1) * pos                        # [H N L]

    # fast softmax using structure of P
    Lambda_gt_0 = Lambda.real > 0                            # [N]
    if Lambda_gt_0.any():
        with torch.no_grad():
            P_max = dt_Lambda * (Lambda_gt_0 * (L-1))       # [H N]
        P = P - P_max.unsqueeze(-1)
    S = P.exp()                                              # [H N L]
    dt_Lambda_neg = dt_Lambda * (1 - 2*Lambda_gt_0)          # [H N]
    # 1 / S.sum(-1) == num / den
    num = dt_Lambda_neg.exp() - 1                            # [H N]
    den = (dt_Lambda_neg * L).exp() - 1                      # [H N]
    W = W * num * reciprocal(den * Lambda)                   # [H N]

    # mixture of softmaxes
    return torch.einsum('hn,hnl->hl', W, S).real            # [H L]

def state_space(u):
    # u: batch of input sequences
    # B: batch size, H: hidden size, L: sequence length
    B, H, L = u.shape
    # compute state space kernel for each of H coordinates
    K = dss_kernel(L)                                       # [H L]
    # multiply two degree L-1 polynomials
    # (u0 + u1*z ... uL-1*z^L-1)(K0 + K1*z ... KL-1*z^L-1)
    # zero-pad them to degree 2L-1 to avoid wrap-around
    K_f = torch.fft.rfft(K, n=2*L)                         # [H L+1]
    u_f = torch.fft.rfft(u, n=2*L)                         # [B H L+1]
    y_f = K_f * u_f                                        # [B H L+1]
    y = torch.fft.irfft(y_f, n=2*L)[..., :L]              # [B H L]
    # yi = ui*K0 + ... u0*Ki
    # residual connection, non-linearity, output projection not shown
    return y
```

Figure 6: Core implementation of DSSSOFTMAX layer (§3.2) in PyTorch.