# OpenReview forum: "Diagonal State Spaces are as Effective as Structured State Spaces"
_NeurIPS.cc/2022/Conference — NeurIPS 2022 Accept_

### Official Review · Reviewer_zKuy · 2022-06-16

**Rating:** 7
**Confidence:** 3
**Soundness:** 4 excellent
**Presentation:** 3 good
**Contribution:** 4 excellent

**Summary:**

The paper builds on structured state space (S4) models and shows that one can achieve similar or slightly better performance with just diagonal state matrices (avoiding the low-rank structure). This is interesting because then the model becomes conceptually simpler and easy to implement. In addition, the paper provides ablations covering initialization and kernel truncation, and an analysis of the learned parameters.

**Questions:**

Questions/suggestions:
- I think the abstract could be largely improved. Some things to consider are: mentioning complexities, mentioning ablation experiments, mentioning analysis, etc.
- Line 17: "denoising objective" --> This is a bit misleading, even more now that denoising diffusion models are in hype. Please find a better terminology.
- Line 39: "DLPR" --> DPLR.
- End of Sec 2: I think it would be better to present here the two views (convolutional and recurrent) instead of just one and then later mentioning the recurrent one.
- Sec. 3: In general, I found it hard to follow with the notation used (and it is not wrong, it is just that it is not presented appropriately). $N$, $H$, and $L$ could be defined beforehand or in Sec 2 (and also give an idea of their relation: like $H<N<L$ if that was the case). I also think that there are some missing "squeeze" and "unsqueeze" operations in the algorithm in the Appendix.
- Algorithm 1: Should there not be an outer product between $\Delta$ and $\Lambda$? Which type of division represents $/$ in line 4? Mention $H=1$?
- Line 137: Isn't $W_{\text{out}}$ the same as $C$ in Eq. 2? (I'm mostly asking for a clarification in the paper)
- Line 152: Any intuition on the initialization of $r$? (same)
- Lines 153-156: So in the end the HiPPO initialization is crucial (according to presented results). Therefore, the model could be considered to still rely on HiPPO theory? More explanation on that would be welcome (in the paper).
- I find there are many differences between the training setups of S4 and DSS (and this can make the reader suspicious). Perhaps it would be instructive to report a dry-run of DSS without those differences (that is, using the S4 setup unaltered) in addition to the reported results.
- Line 192: "images, audio" --> images, and audio.
- Sec 4.1: I think that the section would be much easier to parse if the two ablations were presented one after the other. So first Random Initialization: Question, how, result, and then Truncated Kernels: Question, how, result.
- Line 256: "absolute values" --> Better write "magnitudes"?
- Line 280: Remove "Related work".

Stylistic issues:
- Do not put references in the abstract.
- I am not sure that the way to cite references is correct (not sure about NeurIPS policy here; typically one writes "Surname et al." or just numbers).
- Authors seem to be using "cf." to mean "see". Note that "cf." does not mean "see": https://en.wikipedia.org/wiki/Cf. (that is, cf. should be equivalent to "compare", but the way the authors write it I believe it should be "see").
- A similar thing happens with "provably", where I think they mean "probably". It is not the same and the former implies that it can be proven mathematically: https://wikidiff.com/provably/probably


**Limitations:**

I am happy with the section on limitations.

**Strengths And Weaknesses:**

I like the paper and recommend acceptance. I find it is moderately original, of very good quality, and significant. Clarity can be improved before the final submission.

Strengths:
* The idea is simple and attractive. Simplifications of existing processes is always welcome, especially if they match the accuracy of the previous, more complicated approach.
* I think complexity is reduced. This is perhaps not emphasized enough in the abstract and introduction.
* Good set of evaluations.
* Insightful ablations and analysis.

Weaknesses:
* Presentation. At several points I found that the authors could have made a larger effort in polishing notation or structuring the paper (see some comments below).
* Speed/complexity is only partially assessed. I'd suggest that the authors compare directly with complexities of the original S4 in the small subsection of section 3.2. Alternatively (or in addition), some speed differences can be measured on the same machine.

---

> ### Author Response · Authors · 2022-08-01
> **Author response to Reviewer zKuy**
>
> We thank the reviewer for the encouraging review and several helpful comments! Our response to the reviewer’s comments are as follows:
>
> 1. "DLPR" : fixed  (updated submission attached).
> 2. With Pytorch broadcasting some squeeze/unsqueeze ops can be omitted.
> 3. Algorithm 1: You’re right, as shown in code in the Appendix there is an outer product between $\Delta$ and $\Lambda$ but Algorithm 1 is shown for H=1 (i.e. $\Delta \in \mathbb{R}$) and hence element-wise product suffices.
> 4. C is *not* the same as W_out and they serve different purposes. C is for projecting the N-dimensional states of an implicit state space to a 1-D output y. On the other hand W_out is used for (position-wise) mixing the H outputs of H independent 1-D inputs. It is possible to have H=1 and N=64. W_out is HxH, there’s a C of size N for each of the H coordinates so HxN.
> 5. “/” represents element-wise division.
> 6. HiPPO initialization: Yes indeed our proposed “Skew-Hippo” initialization certainly derives from the HiPPO theory but we agree we do not make it clear in the paper as to why it works. After our work, this question has been answered in a follow-up work with mathematical rigor detailing the connection of Skew-Hippo initialization with the original HiPPO operators We have noted this in the **anonymized footnote 5** in the attached updated version.
> 7. Line 192: fixed.
> 8. Citation style: Neurips policy *allows* “alpha” bibliography style. E.g. following paper from NeurIPS 2021 proceedings uses this style https://papers.nips.cc/paper/2021/file/003dd617c12d444ff9c80f717c3fa982-Paper.pdf (we or our work is in no way related to this paper).
> 9. "cf." : fixed.
> 10. "Provably": the usage is intended to say that “it can be proved” that DSS are as expressive as general SS.

---

### Official Review · Reviewer_riDD · 2022-07-04

**Rating:** 6
**Confidence:** 3
**Soundness:** 3 good
**Presentation:** 2 fair
**Contribution:** 4 excellent

**Summary:**

The paper extends prior work on efficient state space models by simplifying the state transition matrix to be diagonal. The authors make the point that diagonal state space models (DSSMs) are almost as expressive as normal state spaces when considering complex diagonal matrices. This is due to the fact that almost all matrices can be Eigen-decomposed (diagonalized) in the complex plane. The paper introduces 2 parameterizations of DSSMs that are based on this insight and show strong empirical performance on the long-range arena as well as raw audio classification; tasks which require processing very long sequences.

Overall, I think the contributions of this paper a very important for the evolution of the DSSM framework wrt Deep Learning. I have trouble with the presentation of the work though which should be addressed by the authors. It is harder to follow than it should be and the community would benefit from improvements on that end. I am happy to change my score after that accordingly.

**Questions:**

* Background discretization: There is no citation on where this comes from.
* Background: Is the paragraph about computing y and u necessary? It breaks the reading flow and adds mental overhead. Maybe push to the appendix.
* Proposition 1: A is not in C, is it?
* Proposition 1: I fail to see how the resulting K is ensured to be real?
* Initialization of delta_log: Why  is delta_log initialised as e^r with r in U(log(.001), log(.1))? An explanation in the paper would help.
* Initialization of A: Why the eigen-values of only the normal part of the Hippo matrix. What's the reasoning behind this decision?
* Figure 5: I don’t understand what are the delta_log values and how they refer to the real and imag parts of the lambda. Please explain and also extend the caption.
* Figure 5: It is stated that the values of delta_log are larger on short-range tasks? How is this “inline with” larger values modeling long-range dependencies?
* It seems like these models need to be tweaked quite a bit and for a practitioner it would be super helpful to see how brittle these models still are. That is, I would love to see a table/graph of results with non-optimal hparams.



**Limitations:**

Limitations have been addressed.

**Strengths And Weaknesses:**

**Strengths**

* Mostly well written
* very practical insight and simplification
* strong empirical results
* insightful analysis of parameters after training

**Weaknesses**

I only found one weakness but it has to be addressed (IMO):
*Clarity*: I find the main contribution in proposition 1 very hard to follow. This is due to the fact that it is posited without any derivation or intuition, leaving the "elementary" proof for the appendix. I think this is bad practice and puts the burden on the reader to gather all necessary background to understand how the proposition comes together. Given that this is so central to the paper I do not understand the choice of the authors to put the whole derivation of it in the appendix.

Stating the following at least on a high-level before the proposition would make it much easier to understand proposition 1:
1) Most matrices diagonalize over the complex plane (via eigen-decomposition) —> Assume A is one of those —> almost no loss of expressibility.
2) Matrix exponentials are trivial to compute for diagonalised matrices because (VDV^-1)^n = V D^n V^-1. Analogously this is true for power series e^A.
3) V and V^-1 can be subsumed into B and C, show the resulting equation, show that B, C, V, V^-1 can be squashed into a single vector (w).

This would go a long way to make the main paper much more understandable and helping the reader understand where Proposition 1 comes from.

---

> ### Author Response · Authors · 2022-08-01
> **Author response to Reviewer riDD**
>
> We thank the reviewer for the positive review and helpful comments! Our response to the reviewer’s comments are as follows:
>
> 1. We apologize for the lack of proof sketch - we have **included a quick sketch of the proof idea** in the main text as per your suggestion (lines 92-95, footnote 2 in the attached updated submission).
>
> 2. Discretization: added a citation.
>
> 3. Indeed many readers weren’t familiar with FFT-based convolution so we believe this should be the part of the main paper.
>
> 4. Proposition 1: We apologize for the confusion and agree that this subtle point requires clarification. We have changed the statement of Proposition 1 to say that it applies to all kernels over $\mathbb{C}$. As in this work we require the inputs u, outputs y and the kernel K to all be over $\mathbb{R}$ we cast the complex-valued kernel to reals in Step 4 of Algorithm 1, page 4 by simply taking its real part. We have **added a footnote 4 clarifying why this casting does not affect the mathematical soundness of our method and claims**.
>
> 5. Initialization of A: This is a great question and, after our work, this question has been answered in a follow-up work with mathematical rigor detailing the connection of Skew-Hippo initialization with the original HiPPO operators We have noted this in the **anonymized footnote 5** in attached updated version.
>
> 6. Figure 5: We apologize for the confusion. In our implementation we decided to use a separate Delta parameter for the real and imaginary parts of Lambda respectively. I.e. while forming $\Lambda$\*$\Delta$, we formed it as Re($\Lambda$)\*$\Delta$_re + i\*Im($\Lambda$)\*$\Delta$_im instead of Re($\Lambda$)\*$\Delta$ + i*Im($\Lambda$)\*$\Delta$. We have **added a clarification** in the Figure 5 caption and also in the Section A.3 (line 496).
>
> 7. “larger values modeling long-range dependencies”: this was a typo, fixed.

---

### Official Review · Reviewer_NmPw · 2022-07-11

**Rating:** 7
**Confidence:** 3
**Soundness:** 3 good
**Presentation:** 4 excellent
**Contribution:** 3 good

**Summary:**

The paper builds upon the recently proposed S4 architecture, which was shown to be effective at modeling long-range dependencies. The authors propose simplifications to the S4 model by modifying the kernel such that, instead of the originally proposed diagonal-plus-low-rank kernel, they can remove the low rank correction. The theoretical contribution includes proving a proposition which says that under some mild assumptions, this approximation is equivalent to the original kernel formulation. Empirically, the authors show that it is competitive with the original S4 model on the Long Range Arena (LRA) benchmark.

**Questions:**

1. In the last paragraph of Section 2, the authors write that “our idea” is to use an alternate parameterization of state spaces. Isn’t this the original idea in S4?
2. In Section 4.2 (paragraph 4), the authors note that during training, the parameters $\text{Re}(\lambda)$ and $\text{Im}(\lambda)$ move away from the skew-Hippo initialization. Does this mean that this is perhaps not the best initialization for these parameters?


**Limitations:**

The authors have discussed the limitations of their work which is sufficient.

**Strengths And Weaknesses:**

Strengths:
1. The S4 model itself is quite new and interesting, and as such, further studies to analyze and simplify it are useful for the community at large.
2. The 2 methods (DSS-exp and DSS-softmax) are theoretically motivated and demonstrate good empirical performance, where they achieve results comparable to the original S4 on most tasks.
3. The paper is clear and well-written, and the background described in Section 2 is sufficient for new readers (not familiar with S4) to understand the paper.

Weaknesses:
1. My primary concern is that “simplicity” is subjective. The authors motivate DSS by saying that it simplifies computation of the SSM kernel. It would be a stronger result if they can show that it also provides quantitative gains, perhaps in terms of training time/flops.
2. Proposition 1 is based on the assumption that the state matrix $A$ is diagonalizable, and the authors say that this is “a mild technical assumption.” Can they justify this assumption, for instance, by showing that it holds for some toy example where explicit computation of the state matrices is feasible?
3. In Table 1, the large gap between S4 and DSS on the “TEXT” task is attributed to learning rate tuning. Could the authors also try to tune the baseline learning rate for S4, since these numbers can significantly change the “AVG” result?
4. In Section 4.1, the authors try to analyze the effect of restricting DSS to local interactions. As is clear from Table 3, this would be very task-dependent for any model (not just S4), and it is not clear what is the insight gained from this experiment.
5. In Section 4.2, the discussion in the last 3 paragraphs refer to figures in the Appendix. I think the main paper should be self-contained. Perhaps the authors could move the figures to the main paper by reducing some of the “Related work” content in Section 5.

**Update (Aug 8):** After discussions and new time benchmarking results from the authors, I am increasing my rating below.

---

> ### Author Response · Authors · 2022-08-01
> **Author response to Reviewer NmPw**
>
> We thank the reviewer for the positive review and insightful comments! Our response to the reviewer’s comments are as follows:
>
> 1. We believe that simplifying complex approaches and illuminating the main source of the performance improvement of a complex model is essential to research. E.g. DSS is far more accessible to an average ML practitioner and has a much simpler implementation requiring only a handful of lines of code. In particular, compared to S4, the reader no longer needs to know the theory and concepts involving Pade approximations of matrix exponentials (Euler, Bilinear, etc) (2) Woodbury Identity reductions to compute matrix inverse after a low-rank perturbation, and (3) fourier analysis for computing the SSM kernel efficiently via truncated generating functions.
> *EDIT: We **have benchmarked** the running times of DSS-EXP vs S4 and the results are provided in the follow-up comment to Reviewer NmPw below.*
>
> 2. We say that diagonalizability of state matrices is a mild technical assumption as the set of diagonalizable matrices forms a dense subset of the space of NxN matrices over C. I.e. a random NxN matrix, with entries sampled from any large enough subset of C, would have distinct eigenvalues with probability ~1 and hence would be diagonalizable. As a toy example, consider the Fibonacci series F = (0, 1,1,2,3,5,...). This can be computed sequentially via a linear RNN as [F_k, F_{k-1}] = [(1,1), (1,0)].[F_{k-1}, F_{k-2}]. The 2x2 matrix [(1,1), (1,0)] can be diagonalized and its eigenvalues are the golden ratio and its conjugate and hence all F_k’s can be computed in parallel as F_k = c(a^k - b^k) where a, b are the eigenvalues.
>
> 3. We decided to use the recommended configs provided in the official S4 repo as the purpose of our work was not to outperform S4 but to show that the impressive performance of S4 can also be achieved via a far simpler model. On the TEXT task the performance of DSS using the same config as S4 was 76.6 (compared to 75.4 of S4). After our work, researchers have extensively tuned both S4 & DSS-EXP on LRA to improve average test accuracy to ~85 and again found their performance to be within 0.5 points of each other. Some of the changes for this better performance is use of GLU non-linearities instead of GELU, use of cosine learning-rate schedule with warm-up, no use of Dropout, use of weight decay also on “W” params of DSS. More details can be found in this work which we have cited in anonymized footnote 4 in attached updated version.
>
> 4. Section 4.1: The purpose of this experiment is to gain insight into the source of the superior performance of S4/DSS compared to Transformer variants. To demonstrate that the better performance of S4/DSS are due to their ability to better capture long-range interactions, we repeated the experiments after restricting the kernel size as described in the paper and indeed observe a significant reduction in performance, suggesting that S4/DSS indeed capture long-range dependencies. This also highlights that the LRA benchmarking itself has tasks that require long-range reasoning as *if a local model already delivers a good performance on a task then it is not a good task* for testing the long-range abilities of a model in the first place. Therefore, we believe its an informative ablation to include in the paper.
>
> 5. We agree with the reviewer that the main paper should be self-contained. We have included some figures in the main section but unfortunately had to move some of the figures to the appendix due to limited space. We **will bring back** the remaining figures back to the main section in the camera ready version which allows 10 pages of content instead of 9.
>
> 6. Yes, this is also the idea in S4 - the difference being that our parametrization is different compared to S4. We wanted to keep the transition of general SS to DSS self-contained and hence refrained from referring to it here.
>
> 7. This is indeed an intriguing question but unfortunately we do not have anything concrete to say for now except that Skew-Hippo initialization is crucial for empirical performance but it is hard to rule out the possibility of better initializations in the future.

---

> > ### Comment · Reviewer_NmPw · 2022-08-04
> > **Follow-up to the rebuttal**
> >
> > Thanks for responding to each of my comments in detail. Here is my follow-up:
> >
> > 1. I agree that the simplifications in the paper make the theory much more accessible to ML practitioners. However, from an empirical perspective, I still recommend adding results pertaining to any training speed-ups that may be obtained using DSS over the original S4 model.
> >
> > 2. That makes a lot of sense, and I believe it follows from the Lebesgue measure of matrices that are not diagonalizable. A citation would be helpful to convince readers of the validity of this assumption.
> >
> > 3. No further comments.
> >
> > 4. Again, everything the authors have mentioned is true, but it says more about the requirements of the task than the capacity of the model. A transformer, for example, would also perform better with longer context than truncated context, and show similar trends for the different tasks. If the objective here is to show that DSS is better at capturing long range context, perhaps a better comparison would be to show, for increasing values of context, how DSS compares with a Transformer model. If the difference in performance would grow larger as the context size increases, it would conclusively suggest that DSS is better at capturing long range contexts. If computational resources are a concern, perhaps this experiment can be done on a subset of tasks.
> >
> > 5. No further comments.
> >
> > 6. Perhaps it would be useful to make this clarification (through a footnote, for example), to avoid misplaced credit attribution.
> >
> > 7. No further comments.

---

> > > ### Author Response · Authors · 2022-08-08
> > > **Response #2 to Reviewer NmPw**
> > >
> > > **Benchmarking speedups** : For S4, we used the official S4 implementation provided by its authors. Both S4 and DSS-EXP implementations below are based on the [pykeops library](https://www.kernel-operations.io/keops/index.html) for efficiency and benchmarking was performed on a single Nvidia 3090 GPU.
> > > 1. On the Path-X task with input length L=16384, the following are the times on the config provided in the paper (Table 4). For S4 and DSS-EXP all model hyper-parameters are identical.
> > >     - Batch size 1  :   DSS-EXP: 72sec/1000steps    S4: 133sec/1000steps
> > >     - Batch size 16 :   DSS-EXP: 40sec/100steps     S4: 47sec/100steps
> > > For large batch sizes the time taken to perform fft-based-convolution and feedforward part dominates and hence the speedups of DSS are less pronounced. Whereas for smaller batches the time taken to compute the kernel becomes significant and **DSS-EXP provides ~1.8x speedup over S4**.
> > >
> > > 2. We also isolated and benchmarked the time taken to solely compute the S4 kernel vs DSS-EXP kernel (+ perform a backward pass on the sum of all kernel entries). We used H=256, N=64 and varied L. **For large L, DSS-EXP kernel is >2.3x faster than S4 kernel**.
> > > | time (msec) |  L = 4096   |  L = 16384  |  L = 65536  |
> > > | ----------- | ----------- | ----------- | ----------- |
> > > | S4          |    4.0         |   10.7   |    39.8     |
> > > | DSS-EXP     |    4.0         |   6.7    |    17.6     |
> > >
> > >
> > > In line 87, we now clarify that this idea is similar to S4 as per your suggestion.

---

> > > > ### Comment · Reviewer_NmPw · 2022-08-08
> > > > **Closing comment**
> > > >
> > > > Thanks for benchmarking the speed-up. I believe it would be a useful addition to the paper from an empirical perspective. I am happy to increase my original score.

---

> > > > > ### Author Response · Authors · 2022-08-09
> > > > > **Response #3 to Reviewer NmPw**
> > > > >
> > > > > Thank you very much for all the helpful suggestions and for increasing the score!!
> > > > >
> > > > > As per your suggestion we have added the benchmaking results in Section A.4 in the attached updated version.

---

### Meta-Review · Area_Chair_9vYE · 2022-08-20

**Recommendation:** Accept
**Confidence:** Certain

**Metareview:**

This paper proposes a simpler alternative to S4 that achieves comparable performance. The method makes sense and the experiments are thorough. All reviewers agreed this is a good paper. I recommend acceptance.

**Award:**

No

---

### Decision · Program_Chairs · 2022-09-14

Accept